# Research on a Partial Aperture Factor Measurement Method for the AGRI Onboard Calibration Assembly

**DOI:** 10.3390/s22051832

**Published:** 2022-02-25

**Authors:** Xiaolong Si, Xiuju Li, Hongyao Chen, Shiwei Bao, Heyu Xu, Liming Zhang, Wenxin Huang

**Affiliations:** 1Science Island Branch of Graduate School, University of Science and Technology of China, Hefei 230026, China; sixiaolong@aiofm.ac.cn (X.S.); zichen@mail.ustc.edu.cn (S.B.); 2Key Laboratory of Optical Calibration and Characterization, Hefei Institute of Physical Science, Chinese Academy of Sciences, Hefei 230031, China; hychen@aiofm.ac.cn (H.C.); xuhy@aiofm.ac.cn (H.X.); lmzhang@aiofm.ac.cn (L.Z.); 3Shanghai Institute of Technical Physics, Chinese Academy of Sciences, Shanghai 200083, China; lixiuju@mail.sitp.ac.cn

**Keywords:** onboard calibration, partial aperture factor, solar diffuser, absolute radiometric calibration, remote sensors

## Abstract

A partial aperture onboard calibration method can solve the onboard calibration problems of some large aperture remote sensors, which is of great significance for the development trend of increasingly large apertures in optical remote sensors. In this paper, the solar diffuser reflectance degradation monitor (SDRDM) in the onboard calibration assembly (CA) of the FengYun-4 (FY-4) advanced geostationary radiance imager (AGRI) was used as the reference radiometer. It was designed for measuring the partial aperture factor (PAF) for the AGRI onboard calibration. First, the linear response count variation relationship between the two was established under the same radiance source input. Then, according to the known bidirectional reflection distribution function (BRDF) of the solar diffuser (*SD*) in the CA, the relative reflectance ratio coefficient between the AGRI observation direction and the SDRDM observation direction was calculated. On this basis, the response count value of the AGRI and the SDRDM was used to realize the high-precision measurement of the PAF of the AGRI B1~B3 bands by simulating the AGRI onboard calibration measurement under the illumination of a solar simulator in the laboratory. According to the determination process of the relevant parameters of the PAF, the measurement uncertainty of the PAF was analyzed; this uncertainty was greater than 2.04% and provided an important reference for the evaluation of the onboard absolute radiometric calibration uncertainty after launch.

## 1. Introduction

The level of remote sensing data quantification is an important aspect of the advanced technology of remote sensing. The quality of remote sensing data directly affects the correctness of the research directions of the natural sciences and the accuracy of the results. The types, functions and working methods of space remote sensors differ with the rapid development of science and technology and the diverse needs of in-depth research on the earth. The corresponding radiometric calibration methods and technical research have always explored the radiometric calibration schemes of various remote sensors and have continuously improved the radiometric calibration accuracy. In addition to precise and comprehensive measurement calibration before launch, most of the remote sensors have mainly depended on on-orbit alternative calibration and onboard calibration after launch. For onboard calibration, the full aperture all-optical path onboard calibration method was the best choice. However, when the remote sensor had a large aperture or the when available space of the remote sensor was insufficient, onboard calibration methods based on the solar diffuser (*SD*), such as the partial aperture all-optical path and partial optical path full aperture methods, were used. For example, both the advanced baseline imager (ABI) of the Geostationary Operational Environmental Satellite-16 (GOES-16) of the United States [1] and the advanced geostationary radiance imager (AGRI) of the FengYun-4 (FY-4) of China used the partial aperture all-optical path onboard calibration scheme. In fact, the partial aperture onboard calibration scheme reduced the volume of the calibrator by sacrificing the complexity of the onboard calibration physical model. The partial aperture factor (PAF) is one of the key parameters of the AGRI in realizing onboard radiometric calibration. Its measurement method and measurement uncertainty directly affect the function and uncertainty of the AGRI onboard calibration. The partial aperture on-board calibration scheme actually reduced the scaler volume by sacrificing the complexity of the on-board calibration physical model. The measurement of the PAF is one of the key processes to realize the on-board radiance calibration by using this calibration scheme. The improvement of the PAF measurement method and the reduction of measurement uncertainty played a decisive role in reducing the uncertainty of the on-board absolute radiometric calibration of remote sensors.

The AGRI had 6 observation channels in the solar reflective band. It adopts the onboard calibration mode of the partial aperture all-optical path. The calibration assembly (CA) was a passive diffuse reflector that quantitatively scatters sunlight into the AGRI. Unlike the ABI, the CA of the AGRI was equipped with an *SD* stability monitoring device—a solar diffuser reflectance degradation monitor (SDRDM) [1]. The SDRDM consists of integrating sphere and 3 band detectors; it was designed to monitor the degradation of BRDF by measuring the ratio of the reflected radiance of *SD* to sun irradiance. Based on the relative relationship between the characteristics of the AGRI CA and the actual work of the AGRI, this paper proposes a method to measure the PAF of the AGRI calibration optical path by using the SDRDM in the CA as the reference radiometer. First, the SDRDM and the AGRI were used to observe the same radiance source with different energy levels; the numerical relationship between them under the same radiance input was established. Then, the outgoing radiance in the SDRDM observation direction was corrected to obtain the radiance in the AGRI observation direction [2,3,4,5,6] according to the bidirectional reflectance factor (BRF) [7,8,9] of the *SD*. Based on the measured relative relationship between the AGRI and the SDRDM with the full aperture response, according to the observations, the radiance source, the ratio between the calculated full aperture response value of the AGRI and the actual partial aperture measurement response was obtained; then, the PAF of the AGRI was determined. Finally, the PAF measurement equation was analyzed; it was evaluated that the uncertainty of the PAF measured by this method could be better than 2.04%, which provides key parameters for the physical model of the AGRI on-board absolute radiometric calibration after launch. It was also the basis for the evaluation of the uncertainty of the AGRI on-board radiometric calibration.

## 2. Basic Principles

### 2.1. Calibration Principle of the Partial Aperture Based on the SD

The onboard partial aperture all-optical path calibration based on the *SD* mainly limited the radiance flux received by the remote sensor at the calibration time through the partial aperture. Combined with the PAF measured before launch and the radiance response model, the relationship of the full aperture sensor response was obtained with the response count measured by the partial aperture on the satellite. The partial aperture all-optical path marking diagram of the AGRI CA is shown in Figure 1. The AGRI was mainly composed of the *SD*, the SDRDM, partial aperture stop (PAS) and CA box structure. The SDRDM was equipped with three *SD* monitoring bands, 450 nm, 550 nm and 750 nm, which were close to the AGRI measurement band. It can measure the sun for self-calibration, measure the *SD* outgoing radiance and monitor the relative *SD* reflectivity of the three bands [3].

With the known spectral radiance of the *SD*, the response correction coefficient $F\left(B_{j}\right)$ of the AGRI sensor can be expressed as
(1)$$F\left(B_{j}\right)=\frac{k\left(B_{j}\right)L_{SD}\left(B_{j}\right)}{L_{e}\left(B_{j}\right)}$$

In Equation (1), $k\left(B_{j}\right)$ is the PAF of band $\;B_{j}$, $L_{SD}\left(B_{j}\right)$ is the equivalent radiance of band $B_{j}$ output by the *SD*, and $L_{e}\left(B_{j}\right)$ is the radiance of band $B_{j}$ determined by the laboratory radiance response model of the AGRI according to the onboard calibration account. The PAT is set at the front of the AGRI optical path, which had no influence on the AGRI’s own imaging optical path and did not change its optical system parameters. The radiance flux entering the AGRI at the calibration time was mainly considered to be determined by the light passing area of the PAT. That is, the equivalent pupil radiance of the *SD* observed by the AGRI through the PAT relative to the full aperture pupil radiance can be expressed as:(2)$$L_{\mathrm{TP}}\left(\theta_{SD},\phi_{SD};\theta_{v},\phi_{v};B_{j}\right)\;=\;L_{SD}\left(\theta_{SD},\phi_{SD};\theta_{v},\phi_{v};B_{j}\right)\;*\;k\left(B_{j}\right)$$

The *SD* spectral radiance of the AGRI CA in the calibration period can be determined by Equation (3):(3)$$L_{SD}\left(\theta_{SD},\phi_{SD};\theta_{v},\phi_{v};B_{j}\right)\;=\;\cos\theta_{SD,t}\times H_{on-orbit}\left(B_{j},t\right)\times \int_{\lambda_{j_{1}}}^{\lambda_{j_{2}}}\frac{E_{s}\left(\lambda_{j}\right)}{D{\left(t\right)}^{2}}\alpha \left(\lambda_{j},t_{0}\right)\times f_{SD,t}\left(\theta_{SD},\phi_{SD};\theta_{v},\phi_{v};\lambda_{j}\right)\;d\left(\lambda_{j}\right)$$
where $L_{SD}\left(\theta_{SD}{,f}_{SD}{;\theta }_{v}{,f}_{v}{;B}_{j}\right)$ is the *SD* radiance of band $B_{j}$ along the AGRI view at the calibration time; $E_{s}\left(\lambda_{j}\right)$ is the solar spectral irradiance outside the atmosphere [10]; $D\left(t\right)$ is the Sun to Earth relative distance correction factor at the calibration time; $\theta_{SD},\;\phi_{SD}$ are the zenith angle and azimuth angle of the solar incidence in the *SD* coordinate at calibration time, $\theta_{V},\phi_{V\;}$ are the zenith angle and azimuth angle of the AGRI *SD* view, and $\lambda_{j}$ is the wavelength; $f_{SD,t}\left(\theta_{SD},\phi_{SD};\theta_{v},\phi_{v};\lambda_{j}\right)$ is the bidirectional reflection distribution function (BRDF) of the AGRI *SD* view at the calibration time measured in the laboratory; $\alpha \left(\lambda_{j},t_{0}\right)$ is the reflectivity change of the *SD* and is one of the correction factors for the degradation of the *SD* BRDF before launch; $H_{on-orbit}\left(B_{j},t\right)$ is the degradation correction coefficient of the *SD* BRDF of band $B_{j}$ at the time of onboard calibration t determined by the SDRDM long-term monitoring and measurement in the space environment after launch [8,9].

### 2.2. The Measurement of the AGRI PAF

Ideally, the same radiance source provides radiance input for the AGRI imaging optical path and the calibration optical path. The AGRI PAF can be expressed as the ratio of the calibration optical path response count value to the imaging optical path response count value, that is,
(4)$$k\left(B_{j}\right)=\frac{C_{ca,p}\left(B_{j}\right)}{C_{im,f}\left(B_{j}\right)}$$
where *C_im_*(*B_j_*) is the response count value of the imaging optical path of band *B_j_* and *C_ca_*(*Bj*) is the response count value of the calibration optical path of band *B_j_*. The subscript *p* and *f* identify the partial and full aperture. According to the common working mode of the AGRI CA shown in Figure 1, it was difficult to use the same radiance source as the input reference for the imaging optical path and the calibration optical path. After the AGRI was assembled, it was impossible to accurately measure the spectral radiance input in the observation direction of the AGRI both in the imaging optical path and calibration path. Considering that the direction of the SDRDM viewing the *SD* had a fixed geometric relationship with the AGRI viewing the *SD* and that the setting of the SDRDM band was consistent with that of the AGRI, the same type of photoelectric sensors was used in visible and near-infrared bands; they had very close spectral response characteristics. On this basis, a method for measuring the aperture factors of the AGRI based on the relative response relationship between the SDRDM and the AGRI under the same radiance input is proposed:

The linear response relationship between the SDRDM and the AGRI was established by observing the same radiance source at different energy levels. According to the laboratory BRDF measurement data of the *SD*, the relative proportion between the reflectance of the SDRDM observation direction and the AGRI observation direction under the same illumination conditions was established. Under the same illumination angle, the full aperture response count value with the equivalent radiance input of the AGRI was calculated by the count value of the SDRDM observing the *SD* in the CA. The PAF was obtained by comparing the measured value of the AGRI calibration optical path with the full aperture response value in Equation (4). This can be expressed as:(5)$$k\left(B_{j}\right)\;=\;\frac{C\prime_{ca,p}\left(B_{j}\right)}{C\prime_{im,f}\left(B_{j}\right)}=\frac{C\prime_{ca,p}\left(B_{j}\right)\times f_{SD,lab}\left(\theta_{SD},\phi_{SD};\theta_{r},\phi_{r};B_{j}\right)}{\left(a\times C_{\mathrm{SDRDM}}\left(B_{j}\right)+b\right)\times f_{SD,lab}\left(\theta_{SD},\phi_{SD};\theta_{v},\phi_{v};B_{j}\right)}$$
where $C\prime_{ca,p}\left(B_{j}\right)\;$ is the response count value of the *SD* measured by the calibration optical path of the AGRI, $C\prime_{im,f}\left(B_{j}\right)\;$ is the full aperture response count value of the AGRI at the same radiance input equivalent converted according to the outgoing radiance level of the *SD* in the calibration optical path, $C_{\mathrm{SDRDM}}\left(B_{j}\right)$ is the response count value of the SDRDM observing the *SD*, a and b is the conversion relationship coefficient of the response count value for the same radiance source observed by the SDRDM and the AGRI, and $f_{SD,lab}\left(\theta_{SD},\phi_{SD};\theta_{v},\phi_{v};B_{j}\right)$ and $f_{SD,lab}\left(\theta_{SD},\phi_{SD};\theta_{r},\phi_{r};B_{j}\right)$ are the BRDF values of the AGRI observation direction and the SDRDM observation direction under the same lighting conditions, respectively. The flowchart of the PAF measurement is shown in Figure 2.

Before the assembly of the AGRI and its CA, a large aperture integrating sphere was used as a stable radiance reference source. Based on the dynamic range of the AGRI response model, four radiance levels from high to low were set to provide radiance input for the SDRDM and the AGRI at the same time, as shown in Figure 3. The AGRI and integrating sphere radiance source were kept relatively fixed; the relative response relationship between the SDRDM and the AGRI under the same radiance input was established by bringing the SDRDM into and out of the central area of the optical outlet of the integrating sphere radiance source, as shown in Figure 4; the abscissa and ordinate are the response values of the SDRDM and the AGRI observing the same radiance source.

After the assembly of the AGRI with its CA, the light-receiving angle of the AGRI CA at the optimum working distance of the solar simulator light source (SSLS) was adjusted so that its attitude was consistent with the illumination state of incident light when the solar declination angle (the angle between sun vector and XOZ surface) of onboard calibration was 0° (incident along the satellite coordinate system X axis), as shown in Figure 1, in which the solar flux was provided by the SSLS. When the *SD* was illuminated by the SSLS, both the AGRI and the SDRDM began to collect the response value of the *SD* radiance at the same time (the imaging light path was completely blocked to avoid the influence of ambient stray light). The *SD* was not an ideal Lambertian. Under the same illumination angle, the outgoing radiance in the AGRI observation direction was different from that in the SDRDM observation direction. According to the BRDF measured in the laboratory before launch, the radiance measured by the SDRDM was corrected to the outgoing radiance in the AGRI observation direction. The BRDF in the two observation directions is shown in Figure 5. The *SD* had excellent spectral flatness at 400 nm~900 nm. Under the same incidence and observation angle, the relative BRF was approximately 1.0285; the relative change was approximately 0.06%, as shown in Table 1.

According to the relationship between the response count values of the same radiance source being observed by the AGRI and the SDRDM, combined with the measured response count values of the AGRI calibration optical path, the same radiance source observed in the same direction as the AGRI was simulated; the PAF of B1~B3 was obtained according to Equation (5), as shown in Table 2. With the AGRI calibration optical path, the PAF was measured twice. The measurement repeatability of the PAF was better than 0.26%; there were certain differences in the PAF values of various bands due to many factors, such as the placement error between various components of the instrument and the inconsistency of the stray light level.
sensors-22-01832-t001_Table 1Table 1Relative BRF data of different wavelength.ViewersView AngleWavelength (nm)AvgRSTDEV(%)AzimuthZenith400470550650750825AGRI43°53°0.32100.32060.32040.32000.31990.31960.32020.161SDRDM43°53°0.31240.31140.31150.31120.31100.31060.31140.192Relative BRF1.02751.02931.02861.02811.02861.02881.02850.060

## 3. Analysis and Discussion

The AGRI imaging light path was not completely consistent with the calibration light path; the switching between the two paths was mainly completed by adjusting the direction of the AGRI scanning mirror. As shown in Figure 6, considering the environmental stray light and the different areas of the scanning mirror for the incident light of the imaging light path and the calibration light path, the response difference between the imaging light path and the calibration light path with the same radiance source input was not completely determined by the light passing area of the aperture diaphragm. There were many factors affecting the PAF value; the best method for determining it is through a system-level test, but it is very difficult to directly realize the same radiance input in the imaging optical path and calibration optical path. An SSLS with color temperature and ray divergence angle close to those of the sun was used as the irradiance source to obtain the calibration light path response value of the PAF; the response count value of the *SD* was observed with the SDRDM to calculate the response count value that the AGRI imaging light path should have under the same radiance source to indirectly realize a state where the same radiance source provided radiance input for the AGRI calibration light path and imaging light path. Thus, the aperture factor value of the three AGRI visible near-infrared bands was calculated according to the PAF definition.

The PAF was not only the key parameter for realizing onboard absolute radiometric calibration of the partial aperture all-optical path but also the largest source of uncertainty of this onboard calibration method. The uncertainty of some aperture factors measured based on the SDRDM mainly included the following aspects:

The uniformity and stability of the integrating sphere radiance source while determining the relationship between the response values of the SDRDM and the AGRI under the same radiance input affected the uncertainty. The test time of the SDRDM and AGRI was approximately 5 min at every energy level. The integrating sphere radiance source included a halogen lamp light source that had excellent stability, better than 0.25%/30 min, as shown in Figure 7. The radiance surface of the diffuser used in the actual test of the AGRI was approximately the middle area of 1/4 of the light output area of the integrating sphere radiance source; the uniformity of the radiance source could be better than 0.4%. Figure 8 shows the numerical diagram of the nonuniformity of the central area scanned at 70 mm intervals.

The solar simulation light source included a short arc xenon lamp as the illuminant. Affected by gravity and the stability of the power supply, the stability of the spectral irradiance near the working face could reach better than 0.3% within 10 min, as shown in Figure 9. According to the observation of the area of the *SD* and the zenith angle of the light source incident on the *SD* by the AGRI through a partial aperture, it was necessary to evaluate the light source nonuniformity of the irradiance body at ±100 mm from the working face before and after in Φ300 mm; nonuniformity numerical diagrams of three positions between the theoretical working face of the solar simulator and the relative ± 160 mm distance are shown in Figure 10; the nonuniformity of the light source body within this distance range was better than 1.6%.

The direction of the *SD* observed by the SDRDM and AGRI was relatively fixed. Under the same incident angle, the relative change in reflectivity from the *SD* to the SDRDM and AGRI could be better than 0.5% [2,3,4].

The optical fiber of the spectrometer was set along the direction of the AGRI to observe the *SD*, to introduce the solar simulation light source to illuminate the *SD* of the CA to simulate the onboard solar illumination angle, and to simulate the incident light according to the variation range of the illumination angle at the onboard calibration time covering the whole year. The relative difference between the reflected signals of the *SD* in the states with and without the CA box was the proportion of stray light generated inside the box. The stray light contribution of the internal space of the CA to the outgoing radiance of the *SD* was 1%.

According to the B class uncertainty synthesis formula
(6)$$U=\sqrt{\mu_{1}^{2}+\mu_{2}^{2}+\cdots +\mu_{n}^{2}+2\overset{n}{\underset{i=1}{\sum }}\overset{n}{\underset{j=1}{\sum }}\rho_{ij}\mu_{i}\mu_{j}}$$

The uncertainty of the AGRI PAF measurement was calculated, where $\mu_{n}$ is the uncertainty of each parameter determined in the PAF measurement; the correlation coefficient $\rho_{ij}$ between each parameter is assumed to be 0. Uncertainty analysis of the PAF measurement is shown in Table 3.

## 4. Conclusions

The onboard calibration method of the partial aperture all-optical path based on the *SD* was characterized by decreasing the complexity of the calibration system to reduce the volume of the CA, which provided a solution for the onboard absolute radiometric calibration of large aperture remote sensors. In this paper, the implementation principle of the onboard calibration of partial aperture all-optical paths was analyzed in detail; a PAF measurement method was proposed based on the response relationship between the SDRDM and the undetermined calibration remote sensor by testing the same radiance source to simulate the count value of the full aperture response of the imaging path of the undetermined calibration remote sensor. According to this method, the test results of some aperture factors in the B1~B3 bands of the AGRI were given; the test uncertainty of some aperture factors was analyzed in the test process. The uncertainty of this aperture factor test method was approximately 2.04%.

This paper provides a reference for the acquisition method of the PAF parameters of remote sensors using a partial aperture all-optical path on-board calibration scheme before launch, which is of great significance to reduce the volume and weight of the on-board calibrator and to maintain a certain level of radiometric calibration accuracy. In addition, the AGRI had 6 bands in the solar reflective band, while the SDRDM had only 3 monitoring bands, so it could directly measure only the aperture factor values of B1~B3; the corresponding aperture factors could be calculated by the band ratio in other bands. The tests of some aperture factors cannot be decoupled from the remote sensor, so they must be determined by system-level tests. In this paper, some aperture factor tests were carried out for the state of 0° declination incidence on the satellite. Whether the aperture factor changed under different solar incidence angles in actual on-orbit applications had not been tested and verified and needs to be further studied.

## Figures and Tables

**Figure 1 sensors-22-01832-f001:**
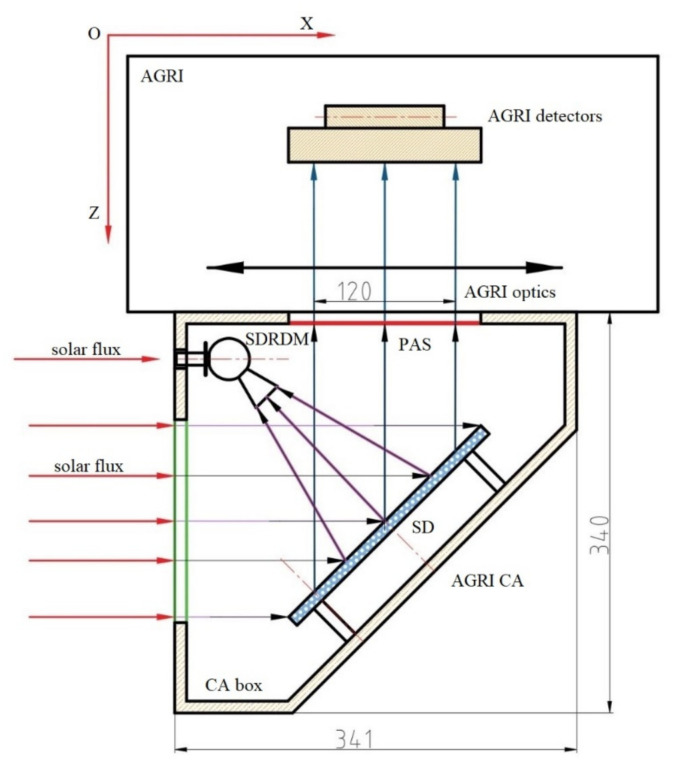
Composition diagram of the AGRI CA.

**Figure 2 sensors-22-01832-f002:**
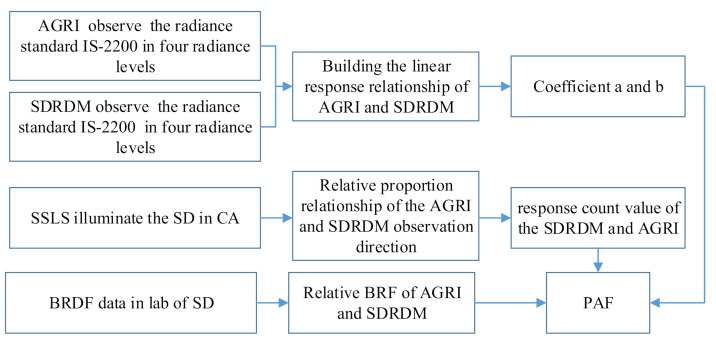
The flowchart of the PAF measurement.

**Figure 3 sensors-22-01832-f003:**
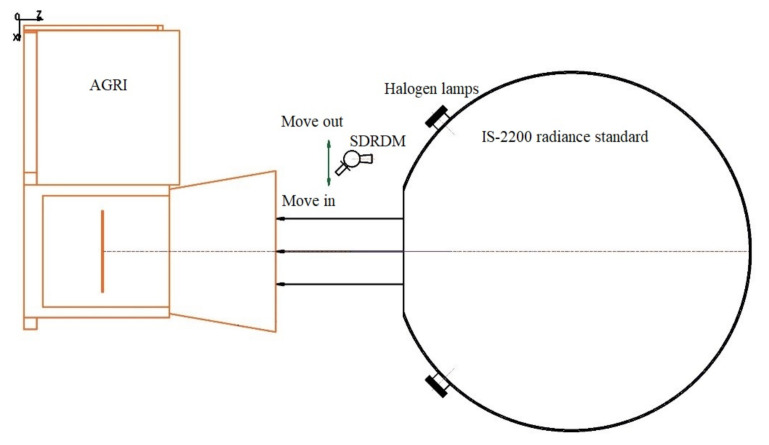
The AGRI and the SDRDM calibration method diagram.

**Figure 4 sensors-22-01832-f004:**
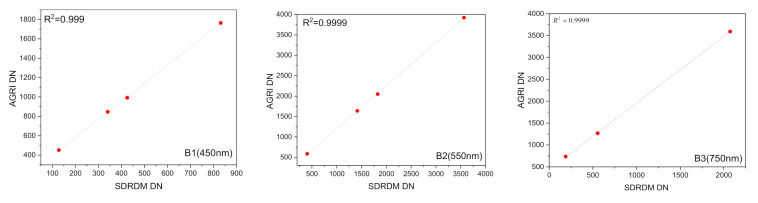
Response relationship of the AGRI and the SDRDM with the same radiance input.

**Figure 5 sensors-22-01832-f005:**
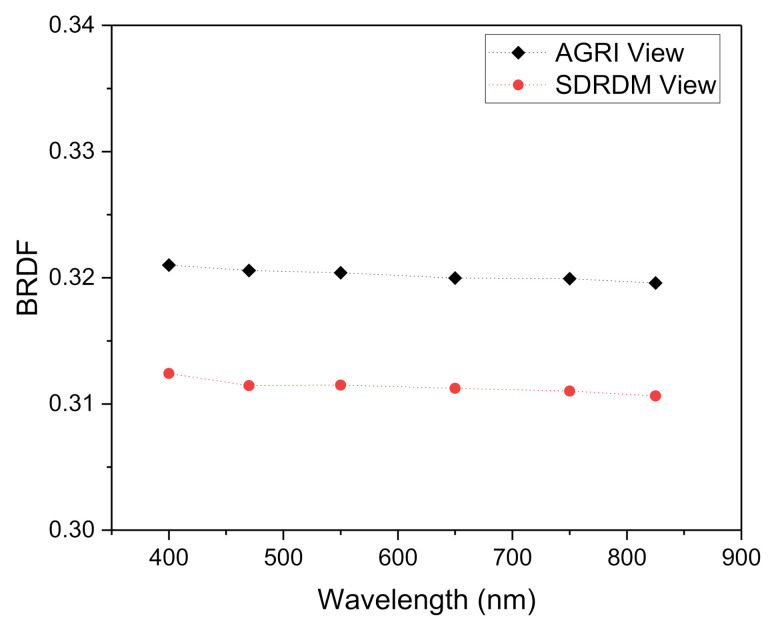
BRDF of the AGRI and SDRDM views with a fixed incident angle.

**Figure 6 sensors-22-01832-f006:**
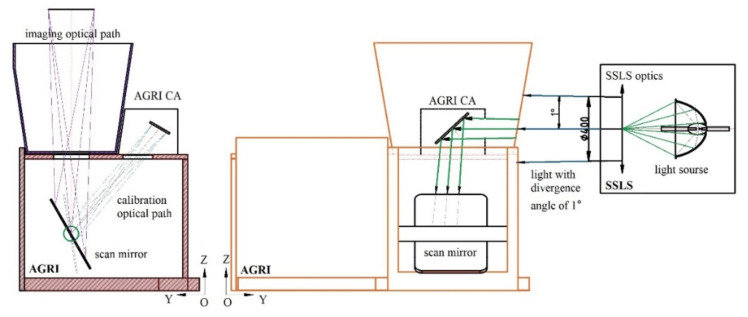
Calibration optical path and imaging optical path of the AGRI.

**Figure 7 sensors-22-01832-f007:**
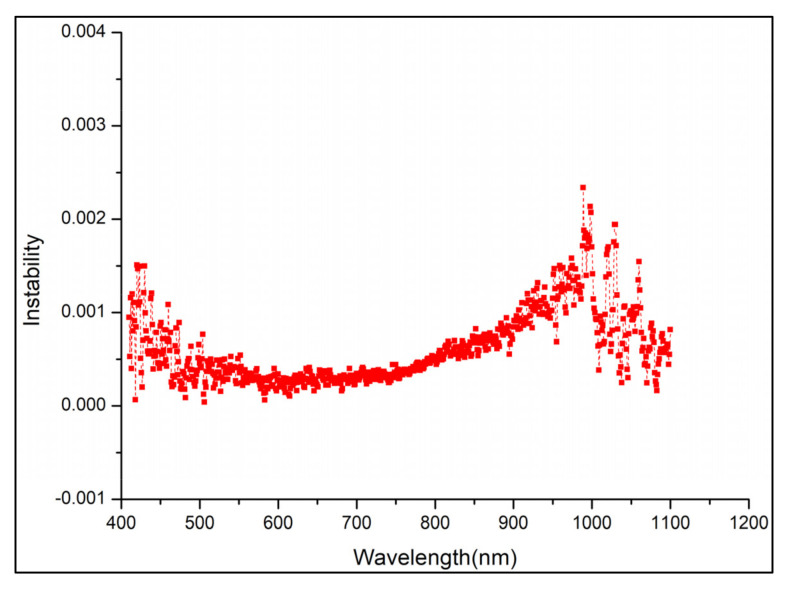
Radiance instability of the integrating sphere in 30 min.

**Figure 8 sensors-22-01832-f008:**
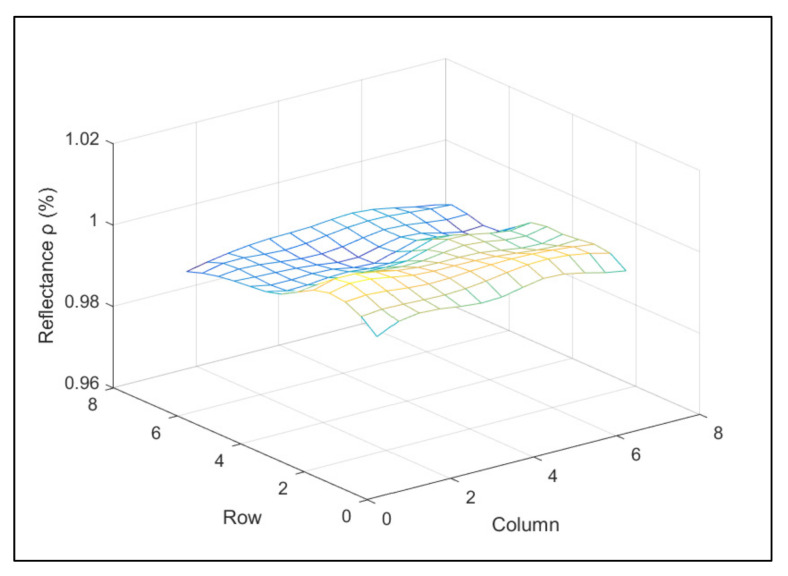
Nonuniformity of the integrating sphere.

**Figure 9 sensors-22-01832-f009:**
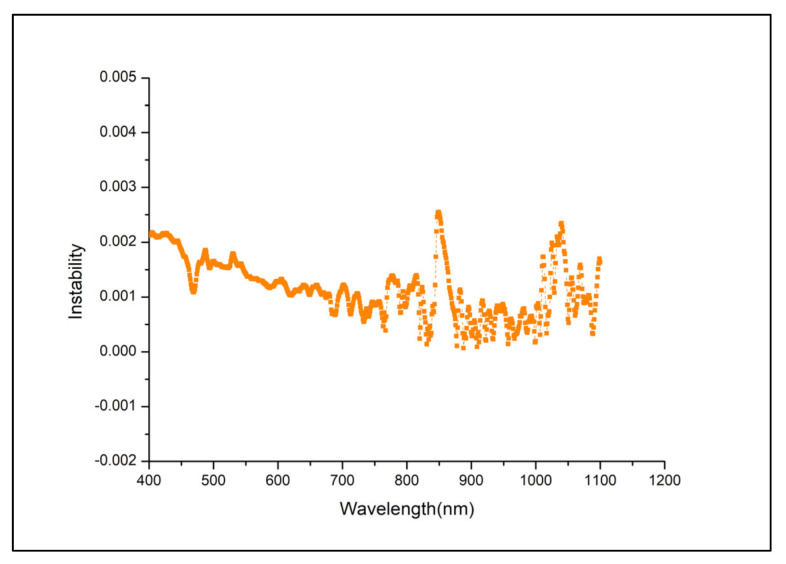
Irradiance instability of the SSLS in 10 min.

**Figure 10 sensors-22-01832-f010:**
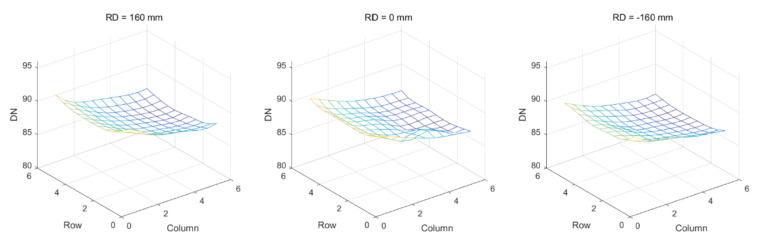
Volume nonuniformity of the SSLS.

**Table 2 sensors-22-01832-t002:** The PAF of the AGRI.

Band	PAF	Repeatability (%)
B1 (450 nm)	0.12011	0.02814
	0.12006	
B2 (550 nm)	0.10940	0.25899
	0.10980	
B3 (750 nm)	0.10346	0.09953
	0.10360	

**Table 3 sensors-22-01832-t003:** Uncertainty analysis of the PAF measurement.

Sources	Uncertainty/%
Stability of the IS radiance source (*μ*_1_)	0.25
Nonuniformity of the IS radiance source (*μ*_2_)	0.40
Stability of the solar simulator source (*μ*_3_)	0.30
Volume nonuniformity of the solar simulator source (*μ*_4_)	1.60
Relative *SD* BRDF measurement (*μ*_5_)	0.50
Stability of the PAF measurement (*μ*_6_)	0.26
Stray light (*μ*_7_)	1.00
Total uncertainty (*U*)	2.04

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
