# Peer review of "Research on a Partial Aperture Factor Measurement Method for the AGRI Onboard Calibration Assembly"

_sensors, 2022, doi:10.3390/s22051832_

Round 1
Reviewer 1 Report
I read the manuscript with great interest. Quite interesting research. I added my comments to the text. However, I would like to point out the following. 1. If this is open information, the reader would be interested in knowing the light diameters of the CA and the size of the sSDRDM. 2. Figure 1 shows the position of the AGRI optics. But figure 5 cannot be compared with figure 1. I would advise you to expand the description by adding a schematic arrangement of elements and their relative position.

Author Response
Response to Reviewer 1 Comments
Point 1: “If we are talking about a geostationary satellite, why the term atmosphere is using?”
Response: The inappropriate description has been deleted.
Point 2: “In my opinion, it is necessary to give a short description of the monitor.”
Response: add the text “The SDRDM consists of integrating sphere and 3 band detectors, it was designed to monitor the degradation of BRDF by measuring the ratio of the reflected radiance of SD to sun irradiance.”
Point 3: “Calibration assembly box structure is absent on the figure. Could you please clarify.”
Response: “The satellite coordinate system description and “CA box” have added in Figure 1, and the relationship between the AGRI and CA has clarified in Figure 1.”
Point 4: “From the point of view of light propagation in the system, what is the function of the scan mirror 1? Is a detector located behind the mirror?”
Response: “There is no detector located behind the mirror. The function of scan mirror 1 is to switch the imaging optical path and calibration optical path. And the problem has clarified in Figure 1.”
Point 5: “Which means p ? Aslo f ?”
Response: add the text “The subscript p and f identify the partial and full aperture.”
Point 6: “Italics?”
Response: These have been modified.
Point 7: “Could you pleace clarify this the phrase? The information is given for 3 bands on the figure 3.”
Response: “it was an omission. And it has been deleted.”
Point 8: “Figure 2”, “Figure 3”
Response: These have been modified in revision.
Point 9: “For a complete understanding, the authors of the manuscript should schematically show the location of the SSLS. It is also not clear further in the text, the definition of angle of onboard calibration=0 deg.”
Response: “The Figure 1 has been showed the illumination relationship between the AGRI CA and SSLS. And the solar declination angle has been explained in the paper.

Reviewer 2 Report
as attachment

Round 2
Reviewer 1 Report
Dear authors.
You have taken note of the comments. However, I made a few more notes on the text of the manuscript. However, I would like to point out the following. You have added additional layout in Figure 1 and Figure 5. Figure 5 is very disorienting to me right now. If you added the bottom layouts, why did you leave the top ? What information does this part give? In figure 5, I made a note. In my understanding, this is a blind ? The thick line, in front of which the rays are indicated, what informational meaning does ? AGRI CA has different sizes on the left and right side of Figure 5. How does the light from the SSLS optics get? The hole is not even indicated on the right side. I strongly recommend that you reconsider the presentation of this figure. For example, make it three-dimensional.

Reviewer 2 Report
The paper can be published in its current form.
Author Response
Thanks for your comment.
Please see the attachment.
